# Characteristic Personality Traits of Multiple Sclerosis Patients—An Unicentric Prospective Observational Cohort Study

**DOI:** 10.3390/jcm10245932

**Published:** 2021-12-17

**Authors:** Eugenia Irene Davidescu, Irina Odajiu, Delia Tulbă, Camelia Cucu, Bogdan Ovidiu Popescu

**Affiliations:** 1Department of Clinical Neurosciences, “Carol Davila” University of Medicine and Pharmacy, 050474 Bucharest, Romania; eugenia.davidescu@umfcd.ro (E.I.D.); delia.tulba@umfcd.ro (D.T.); 2Department of Neurology, Colentina Clinical Hospital, 020125 Bucharest, Romania; irina.odajiu@rez.umfcd.ro (I.O.); cucucameliadaniela@yahoo.com (C.C.); 3Colentina–Research and Development Center, Colentina Clinical Hospital, 020125 Bucharest, Romania; 4Department of Cell Biology, Neurosciences and Experimental Myology, “Victor Babeș” National Institute of Pathology, 030303 Bucharest, Romania

**Keywords:** multiple sclerosis, DECAS, personality profile, quality of life

## Abstract

Background and objectives: Multiple sclerosis (MS) patients tend to present peculiar personality traits that highly impact their quality of life. Our study aimed to determine which personality traits are more common in MS patients compared to a sex- and age-matched control group. Methods and materials: Patients with relapsing–remitting MS along with a sex- and age-matched control group were included. All subjects completed the DECAS Personality Inventory and an additional form including demographic characteristics. Data (including descriptive statistics and univariate and multivariate analysis) were analyzed using SPSS. Results: 122 subjects were included, out of which 61 were in the patient group, mostly females (71.31%) with a mean age of 42.06 ± 10.46 years. Mean duration of disease was 10.18 ± 5.53 years and mean EDSS score was 2.09; 36% of patients were treated with Interferon-beta 1a. Subjects in the patient group presented significantly lower scores for extraversion (*p* = 0.036), specifically those with higher EDSS score, even after adjusting for possible confounders (age, sex, marital status, early retirement, alcohol, and tobacco consumption). Additionally, regarding orientation in life, MS patients were more often philosophers (*p* = 0.001), especially young males, whereas the dominant emotional feeling was less common, the actor profile (*p* = 0.022). Regarding task involvement, MS patients were often passive and compassionate concerning other people. Higher EDSS score also correlated with avoidant (*p* = 0.006) and melancholic (*p* = 0.043) personality traits. Subjects with higher education associated more often pragmatic, experimenter, popular, and optimist traits, whereas the elderly had actor, authoritarian, and experimenter profiles. Conclusions: Some MS patients may have reduced levels of extraversion and specific personality traits compared to age- and sex-matched subjects. Determining the exact personality profile might help the neurologist to establish a better therapeutic alliance and to apply specific interventions.

## 1. Introduction

Multiple sclerosis (MS) is the most frequent demyelinating disease of the central nervous system and affects young females more often [1]. Pathologically, MS is distinguished by multifocal areas of immune-mediated inflammatory demyelination, resulting in loss of oligodendrocytes and astroglial scarring [1]. Since MS lesions can occur anywhere in the central nervous system, their presentation is highly variable. Therefore, MS patients can present various symptoms and signs apart from the classical motor or sensory impairment. Moreover, it was noticed that MS patients tend to present peculiar personality traits, which can serve as conditioning factors for the commencement of a neuropsychiatric disorder, such as depression or anxiety [2]. Psychological peculiarities tend to profoundly alter coping strategies and aggravate the neuropsychiatric conditions [3].

Charcot and Gowers were the first neurologists to describe the affective, mood, and cognitive alterations in patients with MS in the 19th century [3,4]. Charcot accurately described pathological crying and laughing, hallucinations, euphoria, and depression in MS patients [5]. Another feature noticed by Charcot was their strange behavior in opposition to their disability [4], whereas Gowers defined these patients as having a “happy personality” [6]. Current studies also reveal that MS patients have substantially reduced levels of experienced distress and perception of the chaos and helplessness they experience [7]. One possible logical explanation for this behavior could be that temporo-parietal lesions on MRI are associated with emotional blunting [8], higher lesion load as well as cerebral atrophy can determine disinhibition and euphoria [9].

Regarding the five-factor model (FFM) that deals with five core dimensions—neuroticism, extraversion, openness, conscientiousness, and agreeableness [10]—MS patients usually show high neuroticism, loss of empathy, and low agreeableness, which are distinctive features of narcissism and histrionism. According to a study from 2018 [11], the histrionic and narcissistic personality types are frequently encountered in MS patients, along with the avoidant personality described in an earlier study [12].

The personality features described in MS patients could be interpreted as a compensatory mechanism once the diagnosis is established. Due to the heterogeneity of MS symptoms, patients might fear that some of their symptoms could be regarded as conversion traits or overlooked, so they become less agreeable [13]. Multiple sclerosis patients are often viewed as emotionally overreacting, demonstrating increased irritability and tension. Additionally, the attentional type, in particular, displays increased levels of cognitive impulsivity compared to healthy subjects [14]. Apart from that, MS patients are also viewed as less outgoing, appreciative of other people’s thoughts and feelings, and less deliberate [15]. These patients have clear negative self-worth, loss of interest in other people, and thus a reduced comfort in communicating with others [16], determined mainly by concerns regarding the integrity of their body image and functions [17].

Furthermore, the personality traits, mostly neuroticism and extraversion, have a significant impact on MS patients’ quality of life (QoL) [18], apart from worsening physical disability, depression, and self-reported fatigue [19]. The avoidant personality type significantly impacts the QoL, as it appears to be an independent determinant factor of QoL according to one study [20]. Moreover, the personality profile of MS patients influences their treatment adherence [21] and employment status [22]. Therefore, it is important to appreciate MS patients’ personality profiles. This study aimed to determine which personality traits are more common in MS patients compared to a sex- and age-matched control group.

## 2. Materials and Methods

We performed an unicentric prospective observational cohort study, including a group of patients with MS from the outpatient service of the Neurology Department of Colentina Clinical Hospital from Bucharest, Romania—an MS center, part of the National Program for Neurological Diseases—along with a control group of volunteer subjects without neurological disorders. All patients presenting to the outpatient service of the Neurology Department between 18 December 2019 and 10 January 2020 were consecutively included in the study. Inclusion criteria for the patient group consisted of an established diagnosis of relapsing–remitting multiple sclerosis (RRMS) (according to the McDonald criteria [23]) with an evolution exceeding one year and who signed the informed consent, with age ≥ 18 years, whereas in the control group were sex- and age-matched subjects without any neurological comorbidity. No exclusion criteria were applied.

Informed consent was obtained from all the participants. According to our local ethical regulatory items, since our clinic is affiliated to “Carol Davila” University of Medicine and Pharmacy from Bucharest, patients specifically consented to research activities when signing the informed consent upon admission—this being specified in the operational procedure regarding access to archived data in scientific interest—PO MED 01 Edition 1. Rev 0/09.09.2015 and the operational procedure regarding access to patient data, processing, and data protection—PO MED 02 Edition 1. Rev 0/01.07.2019. The study was also approved by the local Ethics Committee (No. 13/17.12.2019) and was performed in accordance with the World Medical Association Declaration of Helsinki from 1975.

All participants from both groups completed the DECAS Personality Inventory [24]. It consists of 97 questions that presume dichotomic (true/false) answers to items evenly assigned to the five fundamental dimensions of personality: openness, extraversion, conscientiousness, agreeability, and emotional stability. Data were introduced and processed using a digital tool that determined the scores and assigned the personality dimension and traits (based on combinations of personality dimensions). The participants also completed a form including questions regarding age, sex, dominant hand, studies, occupation, marital status, religion, alcohol, and tobacco consumption. We also recorded the disease duration, EDSS, and current MS treatment.

Database design and data analysis were performed using IBM SPSS Statistics (version 25.0). Categorical variables were reported as frequency and analyzed with a Chi-square test. Normality was tested with Kolmogorov–Smirnov test. Continuous variables not normally distributed were reported as median (minimum, maximum) and analyzed with the Mann–Whitney U test. Spearman rank-order correlation coefficient was used to measure the strength and direction of association between continuous variables. We adjusted for all the variables significantly associated with the personality dimensions using linear regression, which differed between the two groups (*p* < 0.05). The possible confounders were selected as independent(s) by enter method, with the scales assessing personality dimensions as dependent variables. Hypothesis testing was 2-tailed and statistical significance was defined as *p* < 0.05.

## 3. Results

One hundred twenty-two patients were included in this study, out of which 61 were in the patient group, all Caucasians of Romanian origin. The mean age was 42.06 ± 10.46 years, ranging between 23 and 67 years. Females predominated—71.31% in both groups, which were similar regarding age and sex.

Demographic factors presented in Table 1 were similar between the two groups except for alcohol and tobacco consumption, which were more frequently represented in the control group. Right-handed, married, orthodox, middle-class workers with university degrees, and no tobacco or alcohol consumption prevailed in both groups.

Regarding the patient group, mean disease duration was 10.18 ± 5.53 years, ranging between 1 and 22 years, whereas mean EDSS score was 2.09, with a range between 1 and 7. All patients underwent disease-modifying treatment, Interferon-beta 1a being the most frequently administered treatment (36.06%). Details are presented in Table 2. During the recruitment period, none of the included patients had relapses or an established diagnosis of depression and/or cognitive impairment.

On the DECAS Personality Inventory, the subjects from the patient group obtained slightly higher scores for openness, conscientiousness, and emotional stability, a similar score for agreeability, and a statistically significant lower score for extraversion (*p* = 0.036) in comparison to the control group (Table 3). Extraversion remained associated with MS diagnosis after running the multiple variate analysis for possible confounders such as age, sex, early retirement, marital status, alcohol, and tobacco consumption. Extraversion and openness correlated negatively with EDSS, meaning that patients with higher EDSS values obtained lower scores for extraversion (*p* = 0.043) and openness (*p* = 0.047). Sex correlated positively with age and disease duration (*p* = 0.002). Additionally, older age and religion (*p* = 0.011), specifically being agnostic or atheist, were associated with lower scores of agreeability (*p* = 0.020) in patient group. In this group, higher education levels were positively correlated with openness (*p* < 0.001), conscientiousness (*p* = 0.049), and agreeability (*p* = 0.008). Regarding treatment, Avonex and glatiramer acetate administration were more often associated with a pragmatic personality trait (*p* = 0.010).

Extraversion is a personality dimension involved in several personality profiles according to the DECAS Inventory Profile (24) such as task involvement (along with conscientiousness), dominant emotional feelings (along with emotional stability), interaction with people (together with agreeability), and orientation in life (along with openness). In comparison to the control group, the subjects from the patient group were more often philosophers (*p* = 0.001) and less often actors (*p* = 0.022) (Table 4). Disease duration was not associated with any particular personality profile. Moreover, EDSS correlated positively with the avoidant (*p* = 0.006) and melancholic (*p* = 0.043) personality traits, meaning that patients with higher EDSS scores were more often avoidant and melancholic. In the patient group, male sex was associated more frequently with the philosopher (*p* = 0.039) and melancholic (*p* = 0.039) personality traits. On the other hand, in this group, older age correlated negatively with philosopher (*p* = 0.049) and cautious (*p* = 0.008) traits and positively with actor (*p* = 0.030), authoritarian (*p* = 0.020), and experimenter (*p* = 0.044) profiles. Additionally, in the patient group higher education levels correlated positively with pragmatic (*p* = 0.016), experimenter (*p* = 0.031), popular (*p* = 0.027), and optimist (*p* = 0.004) personality traits and negatively with avoidant (*p* = 0.002), refractory (*p* < 0.001), domestic (*p* = 0.032), actor (*p* = 0.040), and passive (*p* = 0.034) traits. As regards treatment, the only correlation was between the administration of Avonex and glatiramer acetate and pragmatic personality trait (*p* = 0.010).

## 4. Discussion

The results of our prospective observational cohort study revealed that patients with MS exhibit reduced levels of extraversion (*p* = 0.036) compared to the control group, being in line with other publications [15,25]. This difference remained statistically significant even after excluding possible confounders such as age, sex, marital status, early retirement, alcohol, and tobacco consumption. Additionally, patients with higher EDSS values obtained lower scores for extraversion and openness. In combination with other personality dimensions, extraversion influenced essential personality traits such as orientation in life, interaction with people, dominant emotional feelings, and task involvement. Since some MS patients tend to be less extroverted, an empathetic and open attitude could help with communication issues. Interestingly, older patients with higher educational level and lower EDSS values tended to interact better with other people and to have a more positive attitude toward life. They were often experimenters, optimists, and actors because they could adopt efficient coping skills strategies.

Concerning orientation in life, we found that MS patients were more often philosophers, specifically young males, meaning that they generally showed increased interest in inner feelings and ideas. This goes in line with the results of another study in which around 71% of patients had feelings of low self-esteem and of lacking coping skills and competence [26]. Although it is hard to extrapolate this finding to a larger MS population, it is important to have in mind this tendency in some of these patients who could benefit from taking regular psychotherapy sessions.

Concerning interaction with other people, we found that MS patients presented themselves more often as compassionate. This finding suggests that some MS patients are prone to fulfill tasks required of them without arguing or expressing their preferences and rarely have open conflicts with other people. Therefore, others could take advantage of such an attitude. Whenever this is the case, informing their caregivers about their potential vulnerability might protect them from being exploited.

In our study, MS patients presented themselves less often as actors than subjects from the control group regarding the dominant emotional feelings, except for those with higher education levels (*p* = 0.03). This finding draws attention to the potential tendency of some MS patients of not having intense relationships with other people. They might prefer to live more secluded to save themselves from situations where others might observe their deficiencies and avoid others’ attention, thus developing an avoidant personality profile. According to our results, there is an association between this personality type and a reduced level of education. The choice for dysfunctional avoidant coping strategies could be related to cognitive impairment, mainly to the deficits in sustained attention and executive functioning [3]. Suggesting patients to actively participate in different social activities, including those organized by MS patient groups, could significantly improve this issue.

Concerning task involvement, in our study MS patients adopted more frequently a passive attitude, most often subjects with a reduced level of education. Therefore, certain MS patients might be prone to rarely take the initiative and not to respect the deadlines, to lack enthusiasm, and seem indifferent in respect to almost everything. Regarding teamwork, they might seldomly contribute with interventions and lag with their tasks. As stated in another study, such patients tend to be less organized and lack task-focusing, making them susceptible to medication administration errors, missing appointments, loss of employment, relationship dissonance, and other daily problems [26]. We emphasize the need for psychotherapy (especially cognitive-based therapy) in some MS patients who deal with these kinds of problems.

Additionally, this study points to the fact that some MS patients might feel overwhelmed when deep consideration regarding the different treatment options with varying degrees of side effects and benefits is required from them. Therefore, such patients need a high level of instruction, so information must be presented in an accessible manner and not in a single modality but rather in several ways (e.g., verbally, written form, video, diagrams). It is also essential to double-check whether they have sufficiently understood the given material [27].

Furthermore, some MS patients are inclined to have reduced levels of cognitive resources and might use excessive concrete and immature thinking. Due to the fact of this, they usually fuse their feelings and thinking when solving problems, thus tending to operate with less accurate logical systems and admit high ambiguity. Poor self-image and reduced investment in introspection seem to be determined by self-perception and relationship skills [16]. Some MS patients might present a reduced capacity to coordinate internal experiences and emotions since their psychological competence regarding planning and intentional problem-solving, and decision-making strategies are somewhat impaired compared to healthy subjects [28]. As suggested by Kalb et al., apart from periodic screening for cognitive impairment, these patients might benefit from undertaking conceptualized and non-conceptualized treatment as well as physical exercise [29].

Another important observation from our study is that far more often retired subjects were among the patient group than in the control group. Most of these subjects were in early retirement (15 out of retired 16 patients)—a phenomenon also encountered in other reports [30]. Even though more than 90% of patients are employed before a diagnosis is made [31], nearly 70–80% of them cease to be so within five years [30]. Surprisingly, it is estimated that MS disease characteristics, such as difficulty walking, bladder/bowel incontinence, heat sensitivity, fatigue and cognitive impairment, concerning executive functions are responsible for employment loss only in 14–20% of cases [32,33]. Therefore, as other factors had to be blamed for, a study revealed that personality traits significantly influences employment status in MS [22]. However, in our study MS patients had particular personality traits (as described above) independently from early retirement (i.e., early retirement was considered a confounder variable in the multivariate analysis) and higher conscientiousness correlated with higher employment rates. Since it was established that the Symbol Digital Modalities Test (SDMT) could be a crucial predictor of employment status, it could be included in clinical practice in order to help clinicians identify the patients at risk of losing their job and make the required work-related adjustments [22].

An interesting topic relates to the effect of medication on personality traits. In our study, Avonex and glatiramer acetate use was associated with a pragmatic personality. Both drugs have psychiatric side effects (affective and cognitive symptoms) that could alter the personality profiles. Interferons supposedly induce depression mainly by decreased serotoninergic transmission and hyperactivity of the hypothalamic–pituitary–adrenal axis (with high serum cortisol and IL-6 levels) [34]. The mechanisms underlying mania in these patients is less clear but is possibly related to central dopaminergic inhibition [34]. Cognitive impairment seems to be the result of IL-2-enhanced production (which is associated with hippocampal neuronal loss and prevention of long-term potentiation in rats), whereas a relative hyperdopaminergic state underlies the interferon-associated psychotic symptoms [34]. On the other hand, glatiramer acetate might have an antidepressant effect due to the anti-inflammatory and neuroprotective effects [35]. Peripheral administration of glatiramer acetate can enhance the central brain-derived neurotrophic factor (BDNF) activity, release of interleukin-10 and neurogenesis in animal models [35].

The limitations of this study are the following: small sample size and the lack of an objective measure of the personality traits—the questionnaires are completed by the patients, implying a high degree of subjectivity. However, the small number of variables needed for the multivariate analysis certify that MS is independently associated with reduced levels of extraversion (regardless of age, sex, early retirement, marital status, alcohol, and tobacco consumption), albeit the small sample size.

Although there have been other reports exploring personality traits in MS patients, to our knowledge this study is the first one to employ the DECAS Personality Inventory. In opposition to the five-factor model FFM, which explores neuroticism, extraversion, openness, conscientiousness, and agreeableness [10], the DECAS inventory also involves emotional stability, an interesting personality trait that could influence therapeutical compliance. Moreover, the DECAS inventory was designed and validated on a Romanian sample, which we found suitable for our group of patients.

Since MS has such a broad spectrum of clinical manifestations, it is essential to remember that sometimes non-organic features contribute to reduced QoL. Personality traits play a relevant role in how these patients deal with their symptoms and being aware of their exact personality profile eases the neurologist’s interaction with his patients and increases their compliance to treatment. Additionally, by detecting their personality traits, the neurologist can better select the patients who could benefit from specific interventions, including psychotherapy or psychiatric evaluation. It is also very important to mention that the treating physicians should undertake special communication training in order to improve their communication skills and the therapeutic alliance with this group of patients.

## 5. Conclusions

The results of our study support previous evidence that some MS patients exhibit reduced levels of extraversion and, therefore, may be more often philosophers regarding orientation in life and less frequently actors concerning the dominant emotional feelings; they might tend to adopt a passive attitude in respect to task involvement and be more often compassionate with other people. Determining these patients’ personality profile might help the neurologist to establish a better therapeutic alliance and apply specific interventions in order to increase their QoL.

## Figures and Tables

**Table 1 jcm-10-05932-t001:** Demographic parameters.

	Patient Group	Control Group	*p*-Value
**Dominant Fand** - Right - Left			
	58 (95.08%)	57 (93.44%)	0.697
	3 (4.91%)	4 (6.55%)	
**Education** - 8 classes - 10 classes - Vocational school - High school - Post-secondary school - University			
	1 (1.63%)	1 (1.63%)	0.679
	5 (8.19%)	4 (6.55%)	
	2 (3.27%)	4 (6.55%)	
	19 (31.14%)	21 (34.42%)	
	4 (6.55%)	8 (13.11%)	
	30 (49.18%)	23 (37.70%)	
**Occupation** - Retired Early retirement - None - Middle class - Working class			
	16 (26.22%)	5 (8.19%)	0.002
	15 (93.75%)	4 (80%)	
	3 (4.91%)	1 (1.63%)	
	35 (57.73%)	41 (67.21 %)	
	7 (11.74%)	8 (13.11%)	
**Religion** - Orthodox - Atheist - Agnostic - Neoprotestant			
	58 (95.08%)	56 (91.80%)	0.527
	2 (3.27%)	3 (4.91%)	
	0 (0%)	1 (1.63%)	
	1 (1.63%)	0 (0%)	
**Marital Atatus** - Single - Married - Divorced - Widow			
	17 (27.86%)	17 (27.86%)	0.384
	39 (63.93%)	37 (60.65%)	
	2 (3.27%)	6 (9.83%)	
	3 (4.91%)	1 (1.63%)	
**Alcohol Consumption** - None - Occasional - Regular			
	45 (73.77%)	48 (78.68%)	<0.001
	16 (26.22%)	3 (4.91%)	
	0 (0%)	10 (16.39%)	
**Smoking** - Non-smoker - Smoker			
	46 (75.04%)	32 (52.45 %)	0.008
	15 (24.59%)	29 (47.54%)	

**Table 2 jcm-10-05932-t002:** Patient group.

**Disease Duration (years)**	10.18 ± 5.53 (1–22)
**EDSS**	2.09 (1–7)
**Disease-Modifying Treatment** - Interferon beta 1-a Rebif Avonex - Teriflunomide - Natalizumab - Glatiramer acetate - Interferon-beta 1-b - Dimethyl fumarate	
	22 (36.06%)
	12 (19.67%)
	10 (16.39%)
	15 (24.59%)
	12 (19.67%)
	8 (13.11%)
	3 (4.91%)
	1 (1.63%)

**Table 3 jcm-10-05932-t003:** Scores of the DECAS Personality Inventory.

	Openness	Extraversion	Conscientiousness	Agreeability	Emotional Stability
Patient group	51.5	46.9	45.0	50.5	53.0
	(20–80)	(20–80)	(26.7–68.8)	(20–80)	(26.7–73.2)
Control group	48.7	52.7	44.4	50.5	46.4
	(26.7–69.5)	(33.5–81)	(20–62.2)	(31.1–80)	(20–80)
*p*-Value	0.343	0.036	0.308	0.246	0.296

Underline: value is statistically significant.

**Table 4 jcm-10-05932-t004:** Personality profile.

Personality Traits	Patient Group	Control Group	*p*-Value
**Orientation in Life:** - Domestic - Experimenter - Philosopher - Pragmatic			
	18 (29.5%)	21 (34.42%)	0.560
	18 (29.5%)	22 (36.06%)	0.440
	17 (27.86%)	3 (4.91%)	0.001
	8 (13.11%)	15 (24.59%)	0.105
**Interaction with People:** - Compassionate - Competitor - Authoritarian - Popular			
	20 (32.78%)	13 (21.31%)	0.154
	17 (27.86%)	13 (21.31%)	0.400
	11(18.03%)	17 (27.86%)	0.196
	13 (21.31%)	18 (29.50%)	0.298
**Dominant Emotional Feelings:** - Melancholic - Phlegmatic - Optimist - Actor			
	17 (27.86%)	15 (24.59%)	0.681
	18 (29.50%)	11 (18.03%)	0.137
	16 (26.22 %)	14 (22.95%)	0.674
	10 (16.39%)	21 (34.42%)	0.022
**Task Involvement:** - Cautious - Passive - Jovial - Initiator			
	13 (21.31%)	6 (9.83%)	0.081
	22 (36.06%)	20 (32.78%)	0.703
	18 (29.50%)	26 (42.62%)	0.131
	8 (13.11%)	9 (17.47%)	0.794

Underline: value is statistically significant.

## Data Availability

The raw data supporting the conclusions of this article will be made available by the authors without undue reservation.

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
