# Peer review of "Characteristic Personality Traits of Multiple Sclerosis Patients—An Unicentric Prospective Observational Cohort Study"

_jcm, 2021, doi:10.3390/jcm10245932_

Round 1

Reviewer 1 Report

Dear authors,

Your paper explored the peculiar personality traits of MS patients with the aim to establish a better therapeutic strategy and specific interventions. 

My comments are:

In the material and method section:

  1. The authors mentioned only the inclusion criteria for the patient group. Please specify if you applied any exclusion criteria related to already diagnosed depression or cognitive impairment?
  2. Please specify if you included patients having relapses during the study
  3.  Usually the period of the study is specified before the selection criteria of the lots, immediately after specifying the type of study
  4. in table 1: under the heading "Religion" the data in columns 2 and 3 do not correspond to the level of data in column 1 and for "Occupation" p-value is not specified
  5. please change IBM SPSS Statistics (version 25.0) instead of IBM SPSS Statistics 25

In the references section for the first reference some data are missing (journal, year, volume)

Author Response

Reviewer 1:

  1. The authors mentioned only the inclusion criteria for the patient group. Please specify if you applied any exclusion criteria related to already diagnosed depression or cognitive impairment?

Thank you very much for this question. This issue was explained in line 96- we applied no exclusion criteria and included consecutively all the patients who presented to our outpatient section and consented to participate in our study. Nevertheless, we did inquire if they had depression or cognitive decline- please see lines 143-144.

  1. Please specify if you included patients having relapses during the study

The explanation was provided in line 143 (Results section).

  1. Usually the period of the study is specified before the selection criteria of the lots, immediately after specifying the type of study

We have corrected according to your kind suggestion, please see lines 90-92 (Materials and methods section).

  1. in table 1: under the heading "Religion" the data in columns 2 and 3 do not correspond to the level of data in column 1 and for "Occupation" p-value is not specified

Thank you very much for this observation, we have addressed this issue in Table 1 (modifications made with Track changes) (Results section).

  1. please change IBM SPSS Statistics (version 25.0) instead of IBM SPSS Statistics 25

Thank you very much for the suggestion, we have corrected the issue accordingly, please see lines 116-117 (Materials and methods section).

In the references section for the first reference some data are missing (journal, year, volume)

The first reference is an internet source reference and it was cited according to the stated journal rules with the mention of the website and date of access.

Reviewer 2 Report

This manuscript explores personality traits in relapsing-remitting multiple sclerosis patients in Romania, compared with an age- and sex-matched control group. Main findings include reduced extraversion in MS patients. The authors suggest that determining patient personality profiles can help establish a better therapeutic alliance.

While this is an interesting research question, it is not clear what the present study adds to the literature on personality traits in MS patients based on the introduction/background section. Other studies have explored this question (and are cited appropriately); what is new about this study, beyond the Romanian sample?

An additional improvement to the background would be a discussion of treatment effects on personality traits. While this is explored statistically, the reader is not presented with any information, discussion of biological mechanisms, etc.

In the Discussion, MS patients are characterized by multiple traits (e.g., disrespecting deadlines, lacking enthusiasm) but it is not made clear how the present results connect with these fine-grained traits. Also, while the authors do suggest that knowing MS patient personality traits can improve doctor/patient alliances, MS patients are described rather harshly in the Discussion and I wonder if medical providers reading this could come away with a negative impression of MS patients that impacts the alliance for the worse. The authors should re-work the Discussion to highlight specific ways of improving the doctor/patient alliance.

The observed high degree of early retirement in the patient group seems to be a hugely important confounder that deserves more exploration. The authors could compare early retirement patients to other patients to determine the effect of retirement on personality traits.

Author Response

Reviewer 2:

Thank you for your valuable suggestions, they helped us to significantly improve our work. We tried to address them all. Moreover, we made changes in order to improve English writing and Grammar.

While this is an interesting research question, it is not clear what the present study adds to the literature on personality traits in MS patients based on the introduction/background section. Other studies have explored this question (and are cited appropriately); what is new about this study, beyond the Romanian sample?

Thank you for the addressed question. We have provided an explanation in lines 289-295 (Discussion section).

An additional improvement to the background would be a discussion of treatment effects on personality traits. While this is explored statistically, the reader is not presented with any information, discussion of biological mechanisms, etc.

It is a very interesting question for which we are grateful, we tried to provide an eloquent answer in the paragraph between the lines 269-282.

In the Discussion, MS patients are characterized by multiple traits (e.g., disrespecting deadlines, lacking enthusiasm) but it is not made clear how the present results connect with these fine-grained traits. Also, while the authors do suggest that knowing MS patient personality traits can improve doctor/patient alliances, MS patients are described rather harshly in the Discussion and I wonder if medical providers reading this could come away with a negative impression of MS patients that impacts the alliance for the worse. The authors should re-work the Discussion to highlight specific ways of improving the doctor/patient alliance.

Thank you for the proposed issue, we addressed it in the lines 197-198, 205-206, 210-211, 220-222, 230-232, 235-239, 248-251, 296-305. We also reformulated some phrases found in the Discussion topic in order to change this negative impression on MS patients (we only wanted to emphasize the fact that they have particular psychological profiles that deserve attention).

The observed high degree of early retirement in the patient group seems to be a hugely important confounder that deserves more exploration. The authors could compare early retirement patients to other patients to determine the effect of retirement on personality traits.

Thank you very much for this important observation, we addressed this issue in lines 252-268.

Round 2

Reviewer 2 Report

I thank the authors for addressing my concerns on their first draft of the manuscript. Their changes have made this work stronger. However, the authors frequently extrapolate from their personality factor results to characterize MS patients in  a certain way, e.g., "Therefore, they are prone to rarely take the initiative and to not respect the deadlines. They often lack enthusiasm and usually seem indifferent in respect to almost everything." (Lines 505-506). While personality factors can certainly affect taking initiative, etc., I don't believe that painting all MS patients with this "broad brush" is warranted. I would strongly recommend tempering this language by indicating that some MS patients may have problems maintaining deadlines, etc. The authors' goal is to ostensibly improve doctor/patient relationships by highlighting personality factors characteristic of MS that may require more effort from doctors or alternative treatment modalities. However, the authors should be careful not to exacerbate doctor/patient miscommunication by suggesting that all MS patients will act in a certain way. 

Author Response

I thank the authors for addressing my concerns on their first draft of the manuscript. Their changes have made this work stronger. However, the authors frequently extrapolate from their personality factor results to characterize MS patients in  a certain way, e.g., "Therefore, they are prone to rarely take the initiative and to not respect the deadlines. They often lack enthusiasm and usually seem indifferent in respect to almost everything." (Lines 505-506). While personality factors can certainly affect taking initiative, etc., I don't believe that painting all MS patients with this "broad brush" is warranted. I would strongly recommend tempering this language by indicating that some MS patients may have problems maintaining deadlines, etc. The authors' goal is to ostensibly improve doctor/patient relationships by highlighting personality factors characteristic of MS that may require more effort from doctors or alternative treatment modalities. However, the authors should be careful not to exacerbate doctor/patient miscommunication by suggesting that all MS patients will act in a certain way. 

Author's reply to reviewers comments:

We do thank you very much for this valuable suggestion. We rephrased the discussion and conclusions to avoid characterizing all MS patients in this way (please see lines 200-201, 209-211, 213-217, 220-224, 231-233, 240, 247, 252-255, 312, 315). We also made additional English changes, as you required.